# ORATOR: LLM-GUIDED MULTI-SHOT SPEECH VIDEO GENERATION

## ABSTRACT

In this work, we propose a novel system for automatically generating multi-shot speech videos with natural camera transitions, using input text lines and reference images from various camera angles. Existing human video generation datasets and methods are largely centered on faces or half-body single-shot videos, thus lack the capacity to produce multi-shot full-body dynamic movements from different camera angles. Recognizing the lack of suitable datasets, we first introduce *TalkCuts*, a large-scale dataset containing over 500 hours of human speech videos with diverse camera shots, rich 3D SMPL-X motion annotations, and camera trajectories, covering a wide range of identities. Based on this dataset, we further propose an LLM-guided multi-modal generation framework, named *Orator*, where the LLM serves as a multi-role director, generating detailed instructions for camera transitions, speaker gestures, and vocal delivery. This enables the system to generate coherent long-form videos through a multi-modal video generation module. Extensive experiments show that our framework successfully generates coherent and engaging multi-shot speech videos. Both the dataset and the model will be made publicly available. We encourage the readers to view the illustration of the dataset and generated results at https://oratordemo.github.io/.

## 1 INTRODUCTION

Creating multi-shot human speech videos is of significant importance across various industries, including entertainment, education, the film industry, corporate communications, and content creation. The production of such videos involves an intricate interplay of several interacting systems. These systems encompass the way a speaker articulates a given speech, the manner in which the speaker gesticulates and moves within a scene to emphasize certain aspects of the speech and the dynamic camera work that decides between multi-angle shots to emphasize emotions and follow the human subject. However, at its core, a video production begins with a script, and the complex interplaying systems are interpretations of the script, performed and executed by experts such as speakers, educators, comedians, and camera operators, and tied together by a director.

In this work, we pose a novel question: can this intricate process be automated by a system of foundation models? Specifically, can we design foundation models that, given a script and reference images of a person, collaborate to generate a multi-shot human speech video while accounting for all aspects of the production, including vocal delivery, human motion, and dynamic camera work? A high-level overview of this concept is illustrated in Fig. 1.

Recent works have tackled partial aspects of this challenge. Pose-guided methods like AnimateAnyone (Hu, 2024) and MimicMotion (Zhang et al., 2024a) leverage diffusion models to synthesize videos of dancing humans based on driving human pose sequences. Audio-driven methods like EMAGE (Liu et al., 2023) generate 3D proxy geometry from speech inputs. Despite these advancements, current human video generation approaches still fall short in handling the complexity required for multi-shot speech video generation. Firstly, most pose-guided approaches rely on keypoints and images, focusing on domains like dancing (Islam et al., 2019; Guo et al., 2021; Wang et al., 2024a; Xue et al., 2024). These methods often depend on pre-defined keypoints, limiting their ability to function in fully automated systems. While some works attempt to generate speech or talk show scenarios, they are typically constrained to single, static half-body shots(Corona et al., 2024; Zhou et al., 2020), lacking dynamic camera transitions and failing to maintain visual consis-

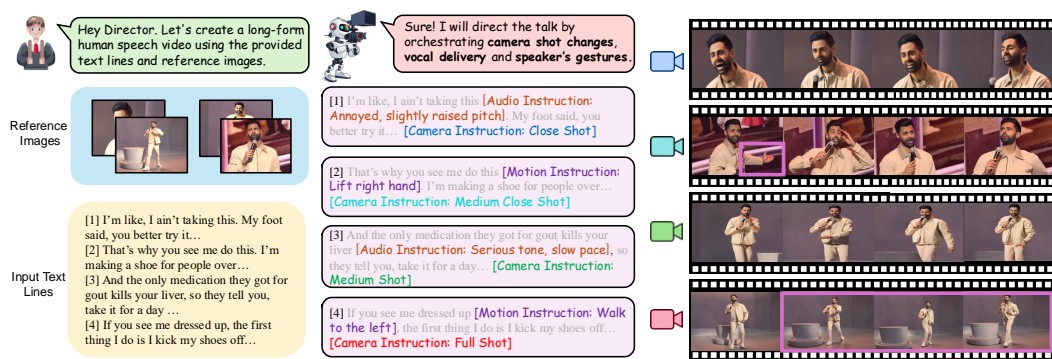

Figure 1: **Multi-shot Human Speech Video Generation**. We propose *Orator*, a fully automated system that generates human speech videos with dynamic camera shots. By organically integrating multiple modules, a DirectorLLM directs camera transitions, gestures, and audio instructions, delivering coherent and engaging multi-shot speech videos.

tency across shots. Secondly, audio-to-gesture methods focus primarily on generating 3D gesture sequences from speech (Wang et al., 2024d; Yi et al., 2023b; Lin et al., 2023), without incorporating these gestures into full video generation or accounting for camera work.

The question remains: how can we generate long-form human speech videos with dynamic camera shots in a holistic end-to-end system? To address this, we propose *Orator*, a pipeline automatically orchestrates the entire process. The system is structured around two key components: a multi-modal video generation module that synthesizes the final video, and a DirectorLLM that guides the generation process. In the multi-modal video generation module, the SpeechGen module first processes the input text and LLM-generated audio instructions to produce synchronized speech audio. Next, the MotionGen module synthesizes 3D motion sequences based on the audio and motion instructions from the DirectorLLM. These 3D motions are projected onto the reference images, following camera shot instructions from LLM to generate 2D keypoint sequences. Finally, the VideoGen module integrates these keypoints with the reference images via a video diffusion model, producing long-form speech videos with natural camera transitions, synchronized gestures, and dynamic vocal delivery. To naturally tie these modules together, a DirectorLLM serves as a multi-role director, guiding the entire system. By providing instructions for camera shot transitions, speaker gestures, and vocal delivery, ensuring that camera angles and actions are synchronized with the speech content and emotional flow. The DirectorLLM determines when to switch between camera angles (e.g., close-up, medium, or wide shots) based on the speech content and emotional flow. Beyond camera control, the DirectorLLM generates natural gesture sequences that align with the speaker's actions and offer vocal delivery guidance to modulate tone, emotion, and pacing. By automating the coordination between these components, our pipeline effectively overcomes the limitations of existing methods, advancing the generation of long-form human speech videos in dynamic settings.

Another key reason why no existing method has holistically addressed this problem is the lack of suitable datasets. Popular public human video generation datasets like TikTok (Jafarian & Park, 2021) and UBC-Fashion (Zablotskaia et al., 2019) focus on dancing and fashion, while datasets like TED Talks (Siarohin et al., 2021) target speech scenarios but are limited in scale and quality. In summary, as shown in Table 1, current human speech video generation benchmarks are restricted by their limited scale, diversity of identities, and lack of comprehensive 2D, 3D, and camera annotations. Moreover, they are constrained to static single-shot settings. To address this gap, we introduce *TalkCuts*, a large-scale dataset specifically curated for human speech video generation with dynamic camera shots. TalkCuts features a diverse collection of videos from talk shows, TED talks, stand-up comedy, and other speech scenarios, comprising over 10,000 unique speaker identities. Each video contains multiple camera shots and is annotated with 2D whole-body keypoints, 3D SMPL-X estimations, and camera trajectories. With 1080p resolution and over 500 hours of footage, *TalkCuts* is the largest public dataset of its kind. All videos have been meticulously filtered and annotated to ensure high quality, providing a comprehensive resource for training and evaluating models capable of generating realistic, multi-shot videos in dynamic speech settings.

Our experimental results demonstrate the effectiveness of our system in generating high-quality speech videos with realistic camera shot transitions. The LLM-directed camera shot planning pro-

duces coherent transitions that align well with the speech content and emotional flow, validating the effectiveness of the LLM's role in guiding both camera shots. The experiments also validate the value of the TalkCuts dataset, showing that it provides sufficient diversity in camera angles, gestures, and speech dynamics to advance high-quality speech multi-shot video generation.

In summary, this paper makes the following contributions: (1) We introduce a novel task of speech video generation with dynamic camera shots across different scales, including head, half-body, and full-body views; (2) We present *TalkCuts*, the first large-scale dataset specifically designed for speech-driven video generation, featuring over 10,000 unique identities, diverse scenarios, and rich annotations including multi-shot camera transitions, 3D SMPLX motion data, and camera trajectories; (3) We propose *Orator*, an automatic pipeline for fine-grained video generation across various speech scenarios, ensuring visual identity consistency. The pipeline integrates a multi-modal generation system guided by a DirectorLLM for camera shot transitions, gesture dynamics, and vocal delivery; (4) Extensive experimental results validate the effectiveness of our approach in generating engaging and realistic multi-shot speech videos.

## 2 RELATED WORKS

**Pose-guided Human Video Generation.** Current research on pose-driven human video generation typically follows a standardized pipeline, with a growing emphasis on efficient pose representations such as skeletons, dense poses, depth maps, mesh models, and optical flow. Early works (Yoon et al., 2021; Chan et al., 2019) predominantly based on GANs (Goodfellow et al., 2020). However, with the development of diffusion model like stable diffusion (SD) and Stable Video Diffusion (SVD), more recent approaches (Tu et al., 2024; Wang et al., 2024b) utilize the UNet structure for video generation. For example, MagicPose (Chang et al., 2023) injects pose features into SD by Control-Net (Zhang et al.) meanwhile MimicMotion (Zhang et al., 2024a) and AnimateAnyone (Hu et al., 2023) extract skeleton poses from targvideo frames using DwPose or OpenPose. Unlike skeleton-based methods, DreamPose (Karras et al., 2023) and MagicAnimate (Xu et al., 2023) employ dense poses, which are directly integrated into the denoising UNet. Furthermore, methods such as Human4DiT (Shao et al., 2024) and Champ (Zhu et al., 2024b) extracts 3D mesh maps using SMPLX.

**Audio-Driven Human Video and Motion Generation** Holistic body motion generation from speech involves synthesizing whole-body motions (Li et al., 2021; Qi et al., 2024). Recognizing that audio signals convey more than just semantic content, (Yi et al., 2023a) propose a method to generate holistic body movements by segmenting the audio signal into different components, each guiding a separate motion generation process. Similarly, learning from masked gesture data, EMAGE (Liu et al., 2023) utilizes four compositional VQ-VAEs for generation. Witnessing the success of diffusion models, more and more works (Chen et al., 2024) began to utilize a diffusion-based structure. MotionCraft (Bian et al., 2024b) exemplifies this trend, using a unified DiT structure to incorporate multimodal controls and achieving state-of-the-art results in audio-to-motion generation. In the domain of audio-driven video generation, preliminary works (Sun et al., 2023; Tian et al., 2024; Ji et al., 2024b) have primarily concentrated on facial regions, ensuring a high degree of consistency between lip movements and the semantic content of the corresponding audio. To expand the generated region, (Corona et al., 2024) synthesizes half-body human videos, while Make-Your-Anchor (Huang et al., 2024c) generates anchor-style full-body videos by translating audio into detailed torso and hand movements using a two-stage diffusion model. ANGIE (Liu et al., 2022) employs an unsupervised feature to model body motion while DiffTED (Hogue et al., 2024) decouples motion from gesture videos while preserving additional appearance information.

## 3 TALKCUTS DATASET

We introduce *TalkCuts*, a large-scale human video dataset specifically designed for speech scenarios such as TED talks and talkshows. *TalkCuts* provides high-resolution speech videos with varying camera shots, and includes diverse modalities such as synchronized texts, audio, 2D keypoints, 3D SMPLX parameters, and camera trajectories, enabling comprehensive multimodal training and evaluation for multi-shot speech video generation. This dataset provides a comprehensive benchmark for future research, facilitating further improvements in speech video generation.

## 3.1 DATA CURATION

**Data Collection.** We performed keyword searches targeting different speech scenarios on YouTube, Xiaohongshu, and Bilibili platforms to crawl copyright-free, high resolution real-world videos. Then, manual filtering was applied to remove low-quality or irrelevant content. Only videos featuring a clearly visible human speaker with corresponding speech audio were retained, while videos with significant obstructions, unclear visuals, or mismatched audio were discarded.

| Dataset | Meta Information | | | | | Modality | | | Camera | |
|---|---|---|---|---|---|---|---|---|---|---|
| | Clips | Frames | Resolution | Hours | ID | 2D Annot. | 3D Annot. | Audio | Trajectory | Shots |
| **Pose-guided Generation Datasets** | | | | | | | | | | |
| TikTok (Jafarian & Park, 2021) | 340 | 93k | 604x1080 | 1.03 | ≈300 | ✗ | ✗ | ✓ | ✗ | Single |
| TED Talks (Siarohin et al., 2021) | 1322 | 197k | 384x384 | - | 173 | ✗ | ✗ | ✓ | ✗ | Single |
| UBC-Fashion (Zablotskaia et al., 2019) | 500 | 192k | 720x964 | 2 | ≈600 | DWPose | ✗ | ✗ | ✗ | Single |
| **Audio-to-gesture Generation Datasets** | | | | | | | | | | |
| Speech2Gesture (Ginosar et al., 2019) | - | - | - | 144 | 10 | OpenPose | ✗ | ✓ | ✗ | Single |
| UBody (Lin et al., 2023) | - | 1051k | - | 11.7 | - | DWPose | SMPL-X | ✓ | ✗ | Single |
| TalkSHOW (Yi et al., 2023b) | 17k | - | - | 38.6 | 4 | ✓ | SMPL-X | ✓ | ✗ | Single |
| BEAT2 (Liu et al., 2023) | - | 32M | 1080P | 76 | 30 | ✓ | SMPL-X | ✓ | ✗ | Single |
| **Ours** | 164k | 57M | 1080P | 507 | 11k+ | DWPose | SMPL-X | ✓ | ✓ | Multi |

Table 1: **Comparison of existing public datasets** for pose-guided video generation (top) and audio-to-gesture generation (bottom), categorized by meta information, modality, and camera details.

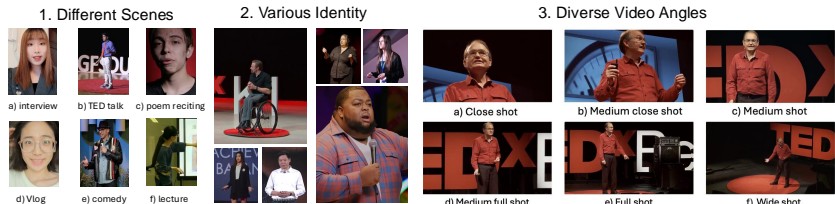

Figure 2: **Visual overview of TalkCuts dataset**. (1) The dataset covers diverse speech scenarios. (2) It features a wide range of identities, capturing individuals of various ethnicities, body types, and ages. (3) Most identities are recorded with multiple diverse camera shots.

**Data Filtering and 2D Keypoint Detection.** We use PySceneDetect (Castellano) to segment each video into multiple clips based on scene transitions. To ensure high-quality clips, we apply RTMDet (Lyu et al., 2022) from MMDetection (Chen et al., 2019) for human detection. Clips are filtered out if no human or multiple humans are detected, or if the bounding box is too small. For the remaining clips, we apply DWPose (Yang et al., 2023a) for human pose estimation to obtain the COCO-whole body pose with 133 keypoints. Final filtering is based on the head keypoint confidence scores, discarding clips with low scores for key facial points.

**Data Statistics.** Our dataset contains over 500 hours of video, with 164K clips and 57M frames, featuring more than 10K unique speaker identities, all in 1080p resolution. Table 1 provides a comprehensive comparison of our dataset with existing speech video datasets, highlighting its scale, diversity, and rich annotations, including multi-camera-shots and 3D SMPLX motion data. Additionally, as shown in Fig. 2, our dataset captures a wide range of speech scenarios (e.g., TED talk, stand-up comedy, presentation, lecture, interview, talkshow and so on), featuring diverse speaker demographics (in terms of race, body type, and age) and various camera shots for each identity, making it suitable for training and evaluating multi-shot speech video generation models.

## 3.2 DATA ANNOTATION

**Camera Shots Definition.** In our paper, we classify camera shots into six types: Close-Up (CU), Medium Close-Up (MCU), Medium Shot (MS), Medium Full Shot (MFS), Full Shot (FS), and Wide Shot (WS) based on established cinematographic principles (as is shown in Fig. 2), as outlined by (Brown, 2016). This classification allows for capturing a wide range of visual details and character interactions, from intimate facial expressions to contextualizing the subject within their environment.

**3D SMPL-X Annotation.** We adopt the SMPL-X (Pavlakos et al., 2019) model to represent 3D human motion. For a given T-frame video clip, the corresponding pose states $\mathcal{P}$ are represented as: $\mathcal{P} = \{\mathcal{P}_f, \mathcal{P}_b, \mathcal{P}_h, \zeta, \epsilon\}$, where $\mathcal{P}_f \in \mathbb{R}^{T \times 3}$, $\mathcal{P}_b \in \mathbb{R}^{T \times 63}$, and $\mathcal{P}_h \in \mathbb{R}^{T \times 90}$ represent the jaw poses, body poses, and hand poses, respectively. $\zeta \in \mathbb{R}^{T \times 10}$ and $\epsilon \in \mathbb{R}^{T \times 3}$ denote the facial expressions and global translation. We initially use the state-of-the-art method SMPLerx (Cai et al., 2024) to estimate the whole-body motion sequence $\mathcal{P}$, but observed limitations in the accuracy of face and hand parameters, specifically $\mathcal{P}_f$, $\zeta$, and $\mathcal{P}_h$. To address this, we refine the hand poses $\mathcal{P}'_h$ using HaMeR (Pavlakos et al., 2024), and improve the jaw poses $\mathcal{P}'_f$ and facial expressions $\zeta'$ using EMOCA (Danecek et al., 2022; Feng et al., 2021). We then combine the refined $\mathcal{P}'_f$, $\zeta'$, and $\mathcal{P}'_h$ into the original pose prediction $\mathcal{P}$ to obtain the final high-quality motion estimation $\mathcal{P}'$.

**Camera Trajectory Annotation.** To reconstruct global camera trajectories from the monocular videos in our dataset, we employ TRAM (Wang et al., 2024c), which builds upon DROID-SLAM (Teed & Deng, 2021) for recovering camera trajectories. To achieve metric-scale accuracy, we refine the estimations by leveraging depth predictions (Bhat et al., 2023) and incorporating semantic cues from the background. This process enables us to recover precise, metric-scale camera motion.

## 4 METHOD

In this section, we present the details of *Orator* for multi-shot human speech video generation. We begin with an overview of the overall system in Sec. 4.1. Following this, we introduce our proposed Multimodal Video Generation Module, composed of the SpeechGen, MotionGen, and VideoGen modules, which sequentially generate the audio, 3D motions, and final video outputs in Sec. 4.2. Then, in Sec. 4.3, we describe the DirectorLLM, which provides instructions for camera transitions, gestures, and vocal delivery to guide the generation process.

### 4.1 OVERALL FRAMEWORK

The overall framework of *Orator* is shown in Fig. 3, which consists of a DirectorLLM and a Multimodal Video Generation Module with a set of specialized generation modules. Given an input speech script $S$ and a set of reference images $\{I_k\}_{k=1}^{K}$ from different camera angles, our framework aims to automatically generate a long-form speech video $V$ with natural camera shot transitions. Firstly, the DirectorLLM takes the speech script $S$ as input and generates camera shot instructions $\{T_i^c\}_{i=1}^{N}$ that determine when and how to transition between shots. These instructions segment the script into $N$ segments $\{S_i\}_{i=1}^{N}$, each corresponding to a distinct shot. For each segment, the DirectorLLM additionally produces motion instructions $\{T_i^m\}_{i=1}^{N}$ for the speaker's gestures and body movements, as well as audio instructions $\{T_i^a\}_{i=1}^{N}$ for vocal delivery, such as tone and pace.

These instructions $\{T_i^c\}_{i=1}^{N}$, $\{T_i^m\}_{i=1}^{N}$, and $\{T_i^a\}_{i=1}^{N}$ are then passed to the corresponding generation modules. The SpeechGen module processes each text segment $S_i$ with the audio instructions $T_i^a$ to generate the vocal output $A_i$. The MotionGen module then takes the generated audio $A_i$ and motion instructions $T_i^m$ to synthesize 3D motion sequences $\{M_i\}_{i=1}^{N}$. Using the camera shot instructions $\{T_i^c\}_{i=1}^{N}$, these 3D motion sequences are projected onto the corresponding reference images to get 2D keypoint sequences $\{K_i\}_{i=1}^{N}$. Finally, the VideoGen module takes the keypoint sequences $\{K_i\}_{i=1}^{N}$ and the reference images $\{I_k\}_{k=1}^{K}$ to generate the final video $V$ via a video diffusion model, incorporating smooth camera transitions, natural gestures, and synchronized audio.

### 4.2 MULTIMODAL VIDEO GENERATION

To enable the automatic generation of long-form speech videos with natural camera transitions, we design a multimodal video generation pipeline. This system integrates three submodules that collaboratively generate synchronized audio, 3D motion sequences, and final video outputs, with instructions provided by the DirectorLLM.

**SpeechGen.** The SpeechGen module is responsible for generating expressive speech audio based on the vocal instructions provided by the DirectorLLM. After receiving the vocal instructions $\{T_i^a\}_{i=1}^{N}$, which specify the tone, pitch, pace, and pauses for each speech segment $S_i$, the SpeechGen module processes the input text lines $S_i$ and generates corresponding audio output $A_i$.

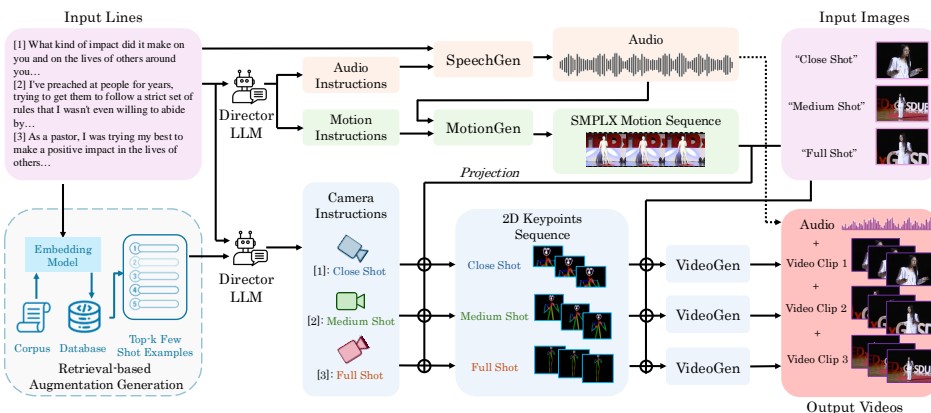

**Figure 3: Pipeline of Orator.** The DirectorLLM processes the input script to generate instructions for camera shots, motion, and audio. These guide the multi-modal generation module to produce the final long-form speech video with natural transitions and gestures.

We utilize the text-to-speech model CosyVoice (Du et al., 2024), which is instruction fine-tuned (Ji et al., 2024a) for enhanced controllability. The model allows for sentence-level adjustments such as emotion, speaking rate, and pitch, as well as token-level controls to insert elements like laughter, breaths, and word emphasis. The SpeechGen module seamlessly integrates these controls from the DirectorLLM, with sentence-level prompts guiding the overall tone and pacing, and tokens like `` for emphasis and `[breath]` for natural pauses. This combined approach ensures that the generated audio synchronizes with the speech content and emotional flow.

**MotionGen.** The MotionGen module generates 3D whole-body motion sequences based on the DirectorLLM's motion instructions $\{T_i^m\}_{i=1}^N$ and the speech audio $A_i$ generated by the SpeechGen module. We leverage MotionCraft (Bian et al., 2024a), a unified diffusion transformer model with multimodal control, to generate SMPL-X 3D motion sequences. The model follows a two-stage coarse-to-fine framework: generating high-level semantic motion from coarse-grained text descriptions and fine-tuning the speech control branches to achieve detailed control over 3D poses.

However, since the pre-trained MotionCraft model was trained only on the BEAT2 dataset (Liu et al., 2023), which limits the variety of generated gestures and movements to a few specific identities, to address this limitation, we freeze the first stage of the model and fine-tune the second stage, specifically the speech control branch, using our dataset. This enables the model to generate more diverse and natural motions tailored to different speech scenarios. Through this process, the MotionGen module generates coherent 3D motion sequences $\{M_i\}_{i=1}^N$ for each segment $S_i$, which are later projected onto the 2D reference images according to the camera shot instructions $\{T_i^c\}_{i=1}^N$ to generate the 2D keypoint sequences $\{K_i\}_{i=1}^N$.

**VideoGen.** The VideoGen module is responsible for generating human speech videos based on the provided reference images $\{I_k\}_{k=1}^K$ and the 2D pose sequences $\{K_i\}_{i=1}^N$ generated by the Motion-Gen module. The goal is to produce videos that not only align with the given pose sequences but also maintain visual fidelity to the reference images throughout the video.

To achieve this, we leverage the pre-trained capabilities of Stable Video Diffusion (SVD) (Blattmann et al., 2023). SVD is known for its performance in generating high-quality, diverse videos from single images, making it an effective model for image-based video generation in our task. By utilizing a pre-trained model, we can significantly reduce the data requirements and computational costs. To enable pose-guide video generation, we integrate ControlNeXt (Peng et al., 2024), a lightweight convolution module for efficient controllable video generation. ControlNeXt efficiently extracts human pose control features using multiple ResNet blocks, which are then integrated into the denoising process of the pre-trained SVD model. Specifically, the conditional control features derived from the pose sequences are added to the denoising branch of SVD at the middle block, allowing the system to directly utilize the pose sequence $K_i$ during video generation. While combining the pre-trained SVD and ControlNeXt models results in smooth video generation aligned with the pose sequences, we observed an issue: the generated faces often lacked fidelity to the reference images. To address this, we fine-tune the ControlNeXt branch on our dataset, specifically adapting the model to the

speech-driven domain. By fine-tuning only the pose control branch, we retain the advantages of the pre-trained SVD model while adapting it to produce videos that maintain consistent visual fidelity with the identity in the reference images. Finally, for each video segment, we generate individual video clips $V_i$ by combining the corresponding 2D keypoint sequence $K_i$ and the reference image $I_{k_i}$. The final long-form speech video $V$ with different camera shots is obtained by concatenating all the generated video clips $\{V_i\}_{i=1}^N$.

## 4.3 DIRETCORLLM AS A MULTI-ROLE DIRECTOR FOR HUMAN VIDEO GENERATION

In this section, we describe the DirectorLLM's role in orchestrating the key elements of video generation: camera shot planning, speaker gesture control, and vocal delivery guidance. Below, we first elaborate how the DirectorLLM handles camera shot planning based on the input speech script.

**LLM as Camera Shots Planner.** The DirectorLLM analyzes the speech script $S$ and generates camera shot instructions $\{T_i^c\}_{i=1}^N$, which are then utilized to segment the script into $N$ segments $\{S_i\}_{i=1}^N$ corresponding to different camera shots. These shot instructions determine the optimal camera angle transitions based on the narrative structure, emotional flow, and key emphasis points within the speech. The LLM selects camera shots based on narrative structure, emotional intensity, and key moments in the script, recommending shot transitions like *"close-up"* (close_up_shot) during emotional highlights and *"wide shots"* (wide_shot) for contextual emphasis. In our approach to automatic shot division, we employ a Retrieval-Augmented Generation (RAG)-based method (Guu et al., 2020; Lewis et al., 2020), leveraging GPT-4o (Achiam et al., 2023) to produce shot transitions $\{T_i^c\}_{i=1}^N$ for video content based on speech. The process begins by extracting text embeddings $E(S)$ from the input speech $S$ using a text-embedding model. We then compute the cosine similarity between the input embeddings $E(S)$ and a pre-computed set of embeddings $\{E(S_j)\}_{j=1}^M$ from our training dataset, the Shot Division Corpus (SDC), which contains speech segments paired with ground-truth shot transitions $\{T_j^c\}_{j=1}^M$. Using this, we retrieve the top-5 most similar speech segments based on the cosine similarity. These retrieved examples $\{S_j\}_{j=1}^5$ and their corresponding shot transitions $\{T_j^c\}_{j=1}^5$, are used as few-shot prompts for GPT-4o (Achiam et al., 2023). Given these contextually relevant examples, GPT-4o generates a shot transition plan $\{T_i^c\}_{i=1}^N$ for the input speech $S$. This approach enables the model to adapt its predictions by learning from past similar examples, effectively capturing the nuanced relationship between speech content and shot division.

**LLM as Motion Instructor.** The DirectorLLM also acts as a motion planner, guiding the speaker's body language, gestures, and movement on stage to enhance the delivery of the speech. For each speech segment $S_i$, the LLM motion instructions $\{T_i^m\}_{i=1}^N$, tailored to the content and emotional tone of the speech. For gestures, the LLM analyzes key points of emphasis and emotion to suggest actions like *"raise right hand"* (gesture_raise_right_hand) or *"open arms"* (gesture_open_arms) during moments of intensity. For more reflective segments, it might recommend subtler movements like *"fold hands"* (gesture_fold_hands). In addition to gestures, the LLM provides instructions for stage movement. Based on the flow of the speech, the LLM suggests where and when the speaker should move on stage, suggesting instructions such as *"move left"* (move_left) or *"step forward"* (step_forward) to maintain a dynamic presence.

**LLM as Voice Delivery Instructor.** The *DirectorLLM* provides fine-grained vocal instructions for intonation, pitch, pace, and emotion, guiding the speaker's delivery to enhance engagement. For each speech segment $S_i$, the LLM generates vocal instructions $\{T_i^a\}_{i=1}^N$ tailored to the emotional tone and context. The LLM could conduct prompt-based control for sentence-level adjustments, controlling overall pitch, emotion, and pacing of a sentence. For example, for introductory remarks or transitions, the LLM might instruct: *"calm tone and lower pitch"* (tone_calm and pitch_low). During critical moments, the LLM can adjust the pace or suggest pauses for emphasis: *"slow down for emphasis"* (slow_pace). The LLM can also leverage token-based control for fine-grained adjustments by inserting word-level emphasis, breathing, or laughter tokens. For instance, it can emphasize key terms: *"The only medication they have for gout kills your liver"* or add realism with [breath] or [laughter] tokens: *"I'm like, I ain't taking this... [breath] My foot said, you better try it."*. By combining these sentence-level and word-level controls, the LLM dynamically adjusts the vocal performance to match the speech's emotional flow, providing a more engaging and natural delivery for speech-driven videos.

## 5 EXPERIMENTS

We evaluate our proposed *Orator* on three key tasks: LLM-guided camera shot transitions, speech-to-gesture generation, and human video generation. Each section presents the metrics and results for these components.

### 5.1 LLM-GUIDED CAMERA SHOT TRANSITIONS

**Metrics.** We assess shot planning accuracy using three key metrics: IoU (Intersection over Union), measuring the overlap between predicted and ground truth shot boundaries (higher IoU indicates better alignment); Accuracy, reflecting the percentage of correctly predicted shot types; and Shot Matching Accuracy (SMA), which evaluates how consistently the predicted shot types match the ground truth at specific time intervals.

| Method | Accuracy↑ | SMA↑ | IOU↑ |
|---|---|---|---|
| Embedding Model | 35.60% | 30.42% | 35.60% |
| Llama 3.1 Z.S. | 20.41% | 23.72% | 10.50% |
| Llama 3.1 R.F. | 24.63% | 44.01% | 13.28% |
| Llama 3.1 RAG | 21.65% | 47.15% | 15.33 % |
| Llama 3.1 Tune | 79.09% | 49.40% | 30.06% |
| GPT-4o Z.S. | 64.34% | 48.34% | 40.59% |
| GPT-4o R.F. | 67.50% | 58.12% | 42.46% |
| GPT-4o RAG (Ours) | 70.66% | 64.06 % | 48.10% |

(a) LLM-guided Camera Shot Transitions

| Method | $FID_H$ ↓ | $FID_B$ ↓ | Face L2↓ | BA↑ | Div↑ |
|---|---|---|---|---|---|
| Talkshow | 129.623 | 143.827 | 11.976 | 7.982 | 5.861 |
| EMAGE | 138.196 | 156.441 | 11.791 | 9.023 | 5.476 |
| MotionCraft | 125.375 | 123.340 | 12.985 | 9.001 | 6.217 |
| Ours | 121.526 | 123.304 | 12.495 | 9.090 | 6.605 |

(b) Speech-to-Motion Generation

Table 2: **Quantitative results of LLM-guided camera shot transitions and speech-to-moton generation.** (a) compares with different baselines designed for camera-shot transition planning; (b) compares with speech-to-motion baselines. best in red and second best in yellow .

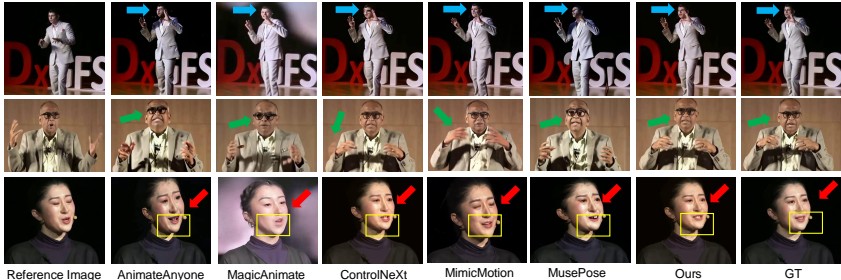

Reference Image   AnimateAnyone   MagicAnimate   ControlNeXt   MimicMotion   MusePose   Ours   GT

Figure 4: **Qualitative Comparison of Human Video Generation Results.** We compare our result with baseline models across close-up, medium, and full shots. Notable artifacts in the baseline models, such as facial distortions, motion blur, or inconsistencies in body movements, are highlighted using arrows and bounding boxes. Our method produces more consistent and realistic results across all shot types, maintaining visual fidelity and smoother transitions compared to the baselines.

**Baselines.** We compare several models: GPT-4o (Achiam et al., 2023), LLaMA 3.1-8B-Instruct (Dubey et al., 2024), and Snowflake-Embed (Merrick et al., 2024). For GPT-4o, we evaluate three setups: RAG-fewshot, random-fewshot, and zeroshot. For the RAG-fewshot setup, we utilized text-embedding-3-small and FAISS (Douze et al., 2024) to retrieve the five most similar examples from the training set to serve as few-shot samples. In contrast, for the random-fewshot setup, we randomly selected five examples from the training set. LLaMA 3.1 (Dubey et al., 2024) is evaluated using similar setups, with additional fine-tuning performed using LoRA (Hu et al., 2021). Snowflake-Embed, being a lightweight embedding model, required the addition of a linear classification head to function as a classifier.

**Result Analysis.** We present the comparison between different baselines in Tab. 2 (a). The embedding model serves as a baseline and shows limited performance without contextual understanding. IoU and SMA values are observed to be better indicators of alignment between the predicted and ground truth shot boundaries compared to accuracy, as high accuracy may due to overfitting. For SMA and IoU, both the Llama (Dubey et al., 2024) and GPT-4o (Achiam et al., 2023) RAG models

outperforms random-fewshot, indicating that selecting relevant examples in our data corpus improves shot planning performance. It is worth noting that the fine-tuned LLaMA model does not achieve a higher IoU than the Embedding Model, but its SMA is significantly better. This suggests that the fine-tuned Llama model has learned some contextual information. On the other hand, GPT-4o (Achiam et al., 2023), although slightly inferior to the fine-tuned Llama (Dubey et al., 2024) in terms of accuracy, shows much higher SMA and IoU, making it the final chosen model.

## 5.2 Speech-to-Gesture Generation

**Metrics and Baselines.** We use $FID_H$, $FID_B$, and Div for quality and diversity measurement. $FID_H$ represents the difference between the hand motion distribution and the ground truth gesture distribution, while $FID_B$ focuses on the distance between the distributions of whole-body motion. Moreover, we use the Beat Alignment Score (Davies & Plumbley, 2007) to measure the alignment between the motion and speech beats and employ L2 Loss to measure the difference between generated and real expressions. We compare our result with the SOTA audio-to-motion methods Talkshow (Yi et al., 2023b), EMAGE (Liu et al., 2023) and MotionCraft (Bian et al., 2024b).

**Comparison on Speech-to-Gesture Generation.** Table 2 (b) demonstrates that our fine-tuned model achieves noticeable improvements across all metrics compared to baseline models. Specifically, our model achieves the best or second-best performance in all metrics, highlighting its effectiveness. By fine-tuning on our dataset, which contains diverse speech scenarios, our model achieves notable improvements over MotionCraft (Bian et al., 2024b), which was originally trained on a limited dataset. This fine-tuning significantly enhances the performance, allowing our model to generate more varied and contextually appropriate gestures, making it suitable for a wide range of speech scenarios.

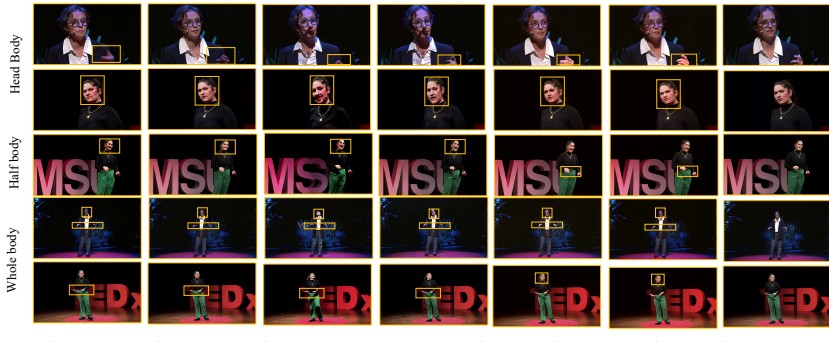

Figure 5: **Ablation Study on Multi-Shot Human Video Generation.** We compares the results of different models after fine-tuning on TalkCuts across various camera shots.

## 5.3 Human Video Generation

| Method | Video Generation Quality | | | | | ID Preser. |
|---|---|---|---|---|---|---|
| | SSIM↑ | PSNR↑ | LPIPS↓ | FID↓ | FVD↓ | ArcFace Dis. ↓ |
| MagicAnimate Xu et al. (2024) | 0.731 | 18.397 | 0.235 | 125.500 | 893.230 | 0.552 |
| Animate Anyone Hu (2024) | 0.754 | 20.468 | 0.176 | 93.230 | 789.360 | 0.450 |
| MusePose Tong et al. (2024) | **0.771** | 19.468 | 0.191 | 106.760 | 823.020 | 0.513 |
| ControlNeXt Peng et al. (2024) | 0.746 | 21.584 | 0.149 | 63.150 | 485.118 | 0.409 |
| MimicMotion Zhang et al. (2024b) | 0.759 | 20.572 | 0.168 | 81.820 | 702.410 | 0.435 |
| Ours | 0.763 | **21.959** | **0.146** | **62.550** | **480.210** | **0.372** |

Table 3: Quantitative Comparison for human speech video generation. Best result is shown in **bold** and the second-best result is shown in underline.

**Metrics.** We assess the generation quality across three dimensions: 1) Single-frame image quality using SSIM (Wang et al., 2004), PSNR (Wang et al., 2004), LPIPS (Zhang et al., 2018), and

FID (Guo et al., 2023); 2) Video quality measured by FVD (Unterthiner et al., 2019); 3) Identity preservation using the ArcFace Distance (Deng et al., 2019).

**Baselines.** We compare our model against previous state-of-the-art methods, including MagicAnimate (Xu et al., 2024), MusePose (Tong et al., 2024), MimicMotion (Zhang et al., 2024a), Animate Anyone (Hu, 2024) (using a third-party implementation[1] due to the original model not being open-source), and ControlNeXt (Peng et al., 2024).

**Evaluation Benchmark.** We provide a test set of 50 video clips from our proposed *TalkCuts* dataset, featuring diverse identities and varying camera shot angles for comprehensive evaluation.

**Result Analysis.** As shown in Table 1, training on our proposed *TalkCuts* dataset with its diverse range of identities and videos featuring dynamic camera shots, our model achieves high scores in both video generation quality and identity preservation. In Fig. 4, we present a qualitative comparison with previous SOTA methods. We observe that previous methods suffer from notable artifacts. For instance, AnimateAnyone (Hu, 2024), MusePose (Tong et al., 2024), and MagicAnimate (Xu et al., 2024) struggle to preserve the human's appearance, generating inaccurate and low-quality facial expressions. Additionally, ControlNext (Peng et al., 2024) produces images with motion blur and misaligned lip movements relative to the speech.

| Method | Video Generation Quality | | | | | ID Preser. |
|---|---|---|---|---|---|---|
| | SSIM↑ | PSNR↑ | LPIPS↓ | FID↓ | FVD↓ | ArcFace Dis.↓ |
| AnimateAnyone | 0.754 | 20.468 | 0.176 | 93.230 | 789.360 | 0.450 |
| AnimateAnyone Tuned | 0.843 | 24.576 | 0.114 | 57.410 | 456.842 | 0.344 |
| MusePose | 0.771 | 19.468 | 0.191 | 106.760 | 823.020 | 0.513 |
| MusePose Tuned | 0.785 | 20.933 | 0.164 | 87.000 | 1014.342 | 0.450 |
| ControlNeXt | 0.746 | 21.584 | 0.149 | 63.150 | 485.118 | 0.409 |
| ControlNeXt Tuned | 0.763 | 21.959 | 0.146 | 62.550 | 480.210 | 0.372 |

Table 4: Quantitative Comparison for ablation study.

**Ablation Study.** To further investigate the effectiveness of our proposed *TalkCuts* dataset, we selected three SOTA methods—MusePose (Tong et al., 2024), Animate Anyone (Hu, 2024), and ControlNeXt (Peng et al., 2024) —and fine-tuned them on our dataset. As is shown in Table. 4, the results show significant improvements across all key metrics after training on our dataset. We also provide further qualitative results from different models across different camera shots in Fig. 5. It is evident that after fine-tuning on our proposed dataset, all models exhibit significantly improved detail in hand and facial features compared to the original results. This enhancement results in more natural and refined body movements and facial expressions under different camera shots, which can be attributed to the high quality and diversity of our dataset. Moreover, we observed distinct behaviors among these models. For instance, Animate Anyone (Hu, 2024), a two-stage diffusion model that first learns appearance and then motion, preserves detailed appearance information well when evaluated frame by frame. However, the generated videos exhibit noticeable temporal instability, resulting in unsmooth motion. In contrast, ControlNeXt (Peng et al., 2024), based on SVD, produces smooth motion across the video but struggles with maintaining facial consistency and identity preservation. Although fine-tuning improved the model's ability to retain appearance details, it still exhibited some discrepancies between the generated faces and the reference images.

# 6 CONCLUSION

In this paper, we introduced a novel framework, *Orator*, for generating human speech videos with dynamic camera shot transitions. Our system integrates an LLM-guided multi-modal generation pipeline, effectively orchestrating the generation of expressive speech audio, natural 3D motion sequences, and coherent video outputs. To address the lack of suitable datasets for this task, we presented *TalkCuts*, a large-scale dataset specifically curated for multi-shot speech-driven video generation, featuring diverse identities, camera shots, and rich annotations. Extensive experiments demonstrate the effectiveness of our approach, advancing the state-of-the-art in speech-driven video generation and opening new avenues for future research in dynamic human video synthesis.

---

[1] https://github.com/MooreThreads/Moore-AnimateAnyone

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

# A  APPENDIX

The appendix is organized as follows:

- Sec. A.1 presents the supplemental website showcasing additional qualitative results;
- Sec. A.2 introduces additional related works;
- Sec. A.3 provides extended quantitative and qualitative results;
- Sec. A.4 gives additional information of proposed TalkCuts dataset;
- Sec. A.5 explains details of RAG process;
- Sec. A.6, Sec. A.7 and Sec. A.8 discuss limitations and future work, potential practical application and potential risks, respectively.

## A.1  ADDITIONAL QUALITATIVE RESULTS

In order to provide more vivid and clear qualitative results, we make a supplemental website demo to demonstrate the *TalkCuts* dataset and the multi-shot human speech video generation results. We encourage the readers to view the results at https://oratordemo.github.io/.

## A.2  ADDITIONAL RELATED WORKS

### A.2.1  HUMAN VIDEO DATASETS.

Recently, various datasets derived from public platforms such as TikTok and YouTube have been introduced to advance human video generation research. For example, the TikTok dataset (Jafarian & Park, 2021) includes 340 short video clips, each lasting 10-15 seconds, primarily featuring dancing humans, while UBC-Fashion (Zablotskaia et al., 2019) consists of 500 fashion-related clips. However, these datasets are limited in both scale and quality. To overcome these limitations, several synthetic datasets (Varol et al., 2017; Patel et al., 2021; Cai et al., 2021; Yang et al., 2023b) have been developed, significantly enhancing the diversity of backgrounds and the scale of training data. For instance, Bedlam (Black et al., 2023) includes thousands of clips with over 1.5 million frames, featuring high-resolution rendered humans in realistic environments.

Recognizing the growing importance of multi-modal data for training, recent datasets Li et al. (2021); Siarohin et al. (2021); Luo et al. (2020) have incorporated various modalities. Furthermore, advancements in annotation tools have facilitated the creation of large-scale, highly realistic datasets.HumanVid (Wang et al., 2024d) consist of more than 50M frames and these frames are well annotated and BEAT2 has more than 32M frames with a high resolution of 1080P. However, these datasets are still limited to identity numbers, which may constrain the ability of generalizations. While datasets like MENTOR (Corona et al., 2024) exists that have over 80k identities dynamic gestures, the dataset remains private. To the best of our knowledge, we are the first public human video datasets that contains thousands of identities.

### A.2.2  MOVIE & CARTOON UNDERSTANDING AND GENERATION

Recent advancements in generative video models have integrated autoregressive frameworks, diffusion models, and large language models (LLMs) to address challenges in long-form, multimodal

video generation and animation. Early methods such as StoryGAN (Li et al., 2019) and Pororo-GAN(Zeng et al., 2019) used GAN-based models for visual storytelling but were limited by contextual inconsistencies in generated frames. To address these limitations, Anim-Director (Li et al., 2024) uses LLMs to autonomously manage the entire animation creation process, refining narratives, generating scripts, and producing contextually coherent animations from brief inputs. Similarly, MovieDreamer (Zhao et al., 2024) combines autoregressive models with diffusion rendering to maintain narrative and character consistency in long-form videos, decomposing complex stories into manageable segments for high-quality visual synthesis. Recent research has also explored using LLMs as "directors" in video generation, where they coordinate various elements similar to a human director managing a film production. (Zhu et al., 2024a) use LLMs to decompose complex prompts into sub-tasks, enabling precise control over 3D scene generation.(Argaw et al., 2022) introduce a benchmark for AI-assisted video editing, focusing on decomposing movie scenes into individual shots based on attributes like camera angles and shot types. This structured representation of shots is conducive to LLM-based systems, which can then manage and edit sequences in a manner similar to a human editor, enhancing the automation of video editing tasks. Additionally, (Rao et al., 2020) propose a subject-centric model to classify shot types, which can enhance LLM-guided video generation by providing structured visual cues. This research suggests that LLMs are well-suited for directing complex video creation processes.

### A.3 Additional Evaluation

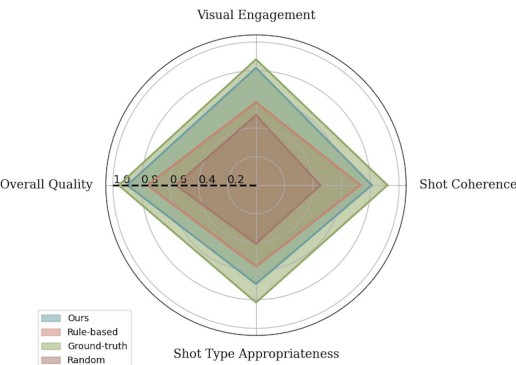

Figure 6: **User study** on camera shot changes directed by LLM.

### A.3.1 Human Evaluation for Camera Shot Changes directed by LLM

For subjective evaluation of camera shot changes generated by the LLM, we conduct an experiment with 20 participants, each rating several criteria on a 1-5 scale (1 = poor, 5 = excellent). We compare our model, ground truth (GT), a rule-based system (shots based on speech length, punctuation, and keywords), zero-shot LLM, and a random baseline (shots randomly assigned). Evaluators will assess from the following aspects:

- Shot Coherence: measures the logical flow between camera shots and evaluates how well the transitions follow the speech content. Evaluators will assess whether the changes in shots are smooth and whether the cuts feel appropriate based on the context. For instance, sharp and abrupt cuts during calm moments would detract from coherence, while fluid transitions during significant speech segments should enhance it.

- Visual Engagement: aimed at evaluating whether the video remains visually captivating and holds the viewer's attention throughout.

- Shot-Type Appropriateness: refers to how suitable the selected shot types (e.g., close-up, medium shot, wide shot) are in relation to the content being delivered. Evaluators will consider whether emotional intensity or important speech moments are reflected with close-up shots and whether wider shots are used to contextualize broader topics or transitions.

- Overall Quality: provides a holistic evaluation of the video, capturing the combined effectiveness of shot selection, transitions, and flow.

The results of the user study are presented in Fig. 6. As demonstrated, our DirectorLLM consistently outperforms the rule-based system, zero-shot LLM, and random baselines in all evaluation criteria. While the ground truth still holds the highest ratings, our model closely approaches its performance, indicating the effectiveness of LLM-driven shot changes and the smoothness of transitions generated by our approach.

### A.3.2 HUMAN EVALUATION ON SPEECH VIDEO GENERATION

To further evaluate the quality of the generated videos, we conduct a user study comparing our results with those from AnimateAnyone, ControlNeXt, and MusePose. The study presents two video clips—one generated by our method and the other by a baseline method (Animate Anyone, ControlNext, or MimicMotion)—to participants. Each participant is asked to evaluate which video they believe demonstrates higher quality, taking into consideration factors such as visual fidelity, smoothness of motion, and consistency in character appearance. We gathered feedback from 20 participants, each of whom evaluated twelve video pairs, with our method compared against each baseline. The results, shown in Fig. 7, highlight a consistent preference for our approach, particularly in terms of maintaining smooth transitions and character consistency. These findings align with our quantitative and qualitative evaluations, supporting the effectiveness of our method in generating high-quality human video synthesis.

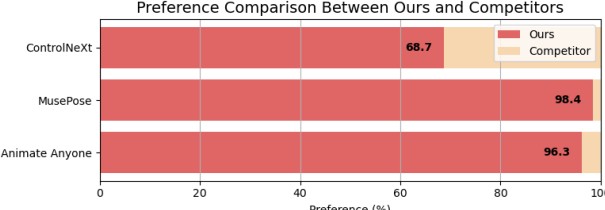

Figure 7: **User study** comparisons on human video generation.

### A.3.3 ADDITIONAL OVERALL EFFECT EVALUATION

| Design | Video Generation Quality | | | Long Video Metrics (↑) | | | | |
|---|---|---|---|---|---|---|---|---|
| | PSNR↑ | LPIPS↓ | FVD↓ | Subject Con. | Background Con. | Temporal Flickering | Motion Smoothness | Imaging Quality |
| w/o. LLM Director | 19.82 | 0.269 | 588.24 | 95.24% | 95.44% | 95.84% | 97.26% | 66.62% |
| w/o. tuned VideoGen | 20.05 | 0.265 | 580.72 | 95.56% | 94.78% | 96.88% | 97.24% | 65.78% |
| Ours | **20.28** | **0.254** | **560.39** | **97.92%** | **96.59%** | **97.24%** | **97.56%** | **68.24%** |

Table 5: **Overall effect evaluation.** Best result is shown in **bold** and the second-best result is shown in underline.

**Overall evaluation.** We assess the generation quality via objective image/video-quality metrics PSNR (Wang et al., 2004), LPIPS (Zhang et al., 2018), and FVD (Unterthiner et al., 2019). Moreover, to better evaluate long videos, we adopt long video metrics based on VBench-Long (Huang et al., 2024b). The quantitative comparison results are presented in Table. 5. For compared baselines, "w/o. tuned VideoGen" denotes that we use the model without tuning the VideoGen module, "w/o. LLM Director" denotes that we remove the LLM Director in generation.

Specifically, for long video metrics, we adopt the following metrics from VBench (Huang et al., 2024a;b): 1) Subject Consistency measures whether the appearance of the subject remains consistent throughout the video. This is assessed using DINO (Caron et al., 2021) feature similarity across frames; 2) Background Consistency evaluates the temporal consistency of background scenes by calculating CLIP (Radford et al., 2021) feature similarity across frames; 3) Temporal Flickering captures imperfections in local and high-frequency temporal consistency. This is measured by taking static frames and computing the mean absolute difference between them; 4) Motion Smooth-

ness focuses on the smoothness of movement rather than the consistency of appearance. Assesses whether motion follows real-world physical laws using motion priors from a video frame interpolation model (Li et al., 2023) and 5) Imaging Quality evaluates frame-level visual quality, such as distortions (e.g., over-exposure, noise, blur), using the MUSIQ image quality predictor (Ke et al., 2021) trained on aesthetic datasets. We follow the evaluation process of VBench Long (Huang et al., 2024b) for long videos. Specifically, we first use PySceneDetect to segment long videos into semantically consistent short clips, ensuring each clip ideally contains no scene cuts. Then the short clips are further divided into fixed-length segments to facilitate slow-fast evaluation. Then for slow branch: we analyze every frame in the short video clip, following VBench's (Huang et al., 2024a) original evaluation method for short videos. For fast branch: we focus on long-range consistency by extracting the first frame from each fixed-length segment and evaluating high-level visual similarity using new feature extractors. Finally, we assess the five metrics (Subject Consistency, Background Consistency, Temporal Flickering, Motion Smoothness, and Imaging Quality) to comprehensively evaluate long video quality.

As shown in Table. 5, our method achieves the best scores across all metrics in Video Generation Quality, indicating superior alignment with real-world videos and higher visual quality. For the long video mertics, the result reveals that removing the LLM Director leads to a significant drop in Subject Consistency and Background Consistency, highlighting the importance of DirectorLLM in coordinating the video generation process. Without tuning the VideoGen module, performance also declines, indicating the necessity of fine-tuning for adapting to speech-driven scenarios.

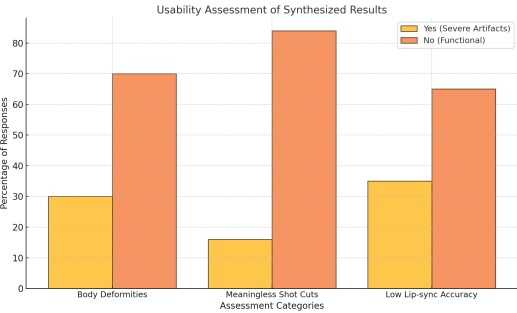

Figure 8: **User study** comparisons on human video generation.

**Additional user study.** To assess the usability of the synthesized results, we conducted a user study involving 15 participants who evaluated 50 randomly generated videos. Each video was assessed on three critical aspects of artifact severity:

- Body Deformities: Distortions or unnatural movements in the generated bodies.

- Meaningless Shot Cuts: Irregular or incoherent transitions between shots.

- Low Lip-sync Accuracy: Mismatches between speech and lip movements.

Participants provided binary feedback ("Yes" or "No") for each aspect, indicating whether severe artifacts were present. We demonstrate the results in Fig. 8. The chart highlights the proportion of "Yes" (severe artifacts) and "No" (functional) responses for each category. The study reveals distinct patterns in the usability of synthesized results: Low Lip-sync Accuracy emerged as the most significant challenge, with 35% of the results exhibiting severe artifacts. This suggests room for improvement in synchronizing speech with facial animations. Body Deformities were noted in 30% of responses, indicating a need for enhanced robustness in body generation, particularly to avoid unnatural or distorted poses. Meaningless Shot Cuts, with a lower artifact rate of 16%, indicate relatively better performance in maintaining coherent transitions, though further optimization is desirable. These findings underline the importance of addressing lip-sync accuracy and body generation robustness to improve the overall usability of synthesized videos. The relatively lower issues with shot cuts suggest that the system's shot planning module is more reliable but still warrants refinement to minimize occasional artifacts.

| Design | Video Generation Quality | | | Long Video Metrics (↑) | | | | |
|---|---|---|---|---|---|---|---|---|
| | PSNR↑ | LPIPS↓ | FVD↓ | Subject Con. | Background Con. | Temporal Flickering | Motion Smoothness | Imaging Quality |
| LLM-combined | 18.24 | 0.342 | 712.42 | 91.18% | 92.78% | 94.98% | 96.68% | 62.24% |
| End-to-end | 17.65 | 0.356 | 808.19 | 90.24% | 91.18% | 93.29% | 95.79% | 61.79% |
| Ours | **20.28** | **0.254** | **560.39** | **97.92%** | **96.59%** | **97.24%** | **97.56%** | **68.24%** |

Table 6: **Evaluation on end-to-end systems.** Best result is shown in **bold**.

#### A.3.4 ADDITIONAL EVALUATION ON END-TO-END SYSTEM

In this section, we evaluate alternative end-to-end system designs and compare them with our proposed modular pipeline. Table. 6 presents the results for the following methods: 1. "End-to-end": A direct approach where the VideoGen module is fine-tuned end-to-end on our dataset using text and a reference image as input; 2. "LLM-combined": A design inspired by works such as (Hong et al., 2023; Dong et al., 2023; Zhen et al., 2024; Xiang et al., 2024), where the DirectorLLM is integrated with the video generation model, directly guiding the diffusion process by providing contextual features.

From Table. 6, we observe that our proposed pipeline outperforms both end-to-end designs across all metrics, including PSNR, LPIPS, and FVD, as well as long video metrics like Subject Consistency, Motion Smoothness, and Imaging Quality. The "End-to-end" design struggles with maintaining high fidelity and temporal consistency, leading to lower scores across metrics. This indicates the challenges of learning all aspects of video generation in a unified model, particularly under limited data conditions. The "LLM-combined" approach achieves better results than the direct end-to-end model but still falls short of our modular design. This highlights the difficulty of integrating multi-modal controls (e.g., camera shots, motion, and audio) into a single end-to-end framework without loss of interpretability and control. These results validate our choice of a modular pipeline, where the DirectorLLM orchestrates specialized generation modules for SpeechGen, MotionGen, and VideoGen. The modular approach provides: 1) Interpretability: Each submodule's output can be analyzed and optimized independently and 2) Flexibility: Components like VideoGen can be fine-tuned separately to adapt to domain-specific requirements.

While extending to end-to-end designs is a promising direction, particularly with access to larger and higher-quality datasets, our modular pipeline serves as a strong baseline for this challenging task. It lays the groundwork for future research into more unified systems.

#### A.3.5 ADDITIONAL QUALITATIVE RESULTS

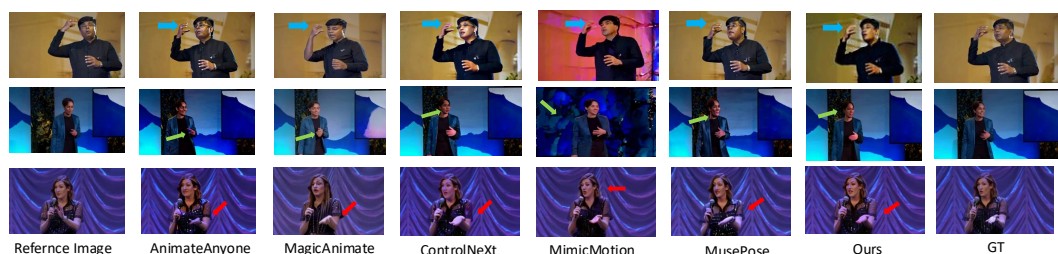

Refernce Image    AnimateAnyone    MagicAnimate    ControlNeXt    MimicMotion    MusePose    Ours    GT

Figure 9: **Additional Qualitative Comparison of Human Video Generation Results.** We include additional examples to compare our results with baseline models. Key artifacts in the baseline models, such as facial distortions, motion blur, and background inconsistencies, are highlighted with arrows. In contrast, our method delivers more consistent and realistic outputs, preserving visual fidelity and achieving smoother transitions compared to the baselines.

As shown in the Fig. 9, given a reference image and the corresponding pose, we test multiple baseline models alongside our proposed model. Existing baseline models exhibit issues such as facial distortions, hand deformities, and background inconsistencies. In contrast, the results generated by our method are closest to the ground truth, with significant improvements in facial and hand details, as well as better background consistency. This further illustrates that the data diversity we provide enhances model performance, showcasing the effectiveness of both our dataset and method.

## A.4 ADDITIONAL INFORMATION ON TALKCUTS

### A.4.1 MANUAL SCREENING PROCESS

We outline the detailed steps for ensuring consistency and accuracy in the manual screening process:

1. **Scene Segmentation Validation:** After performing automated scene segmentation using PySceneDetect, human reviewers verify the correctness of the detected shot boundaries. Reviewers ensure that transitions occur at logical points, such as changes in subject focus or significant shifts in speech content. Incorrectly segmented scenes are manually adjusted to improve coherence.

2. **Subject Quality Evaluation:** Each video clip is manually inspected to evaluate the clarity and quality of the human subject within the frame:

   - *Clarity:* The subject must be clearly visible without blurring or obstructions.
   - *Lighting:* The subject's features must be well-lit and distinguishable.
   - *Framing:* The subject must be proportionally centered in the frame.

Clips failing to meet these criteria are discarded.

3. **Consistency and Annotation Accuracy:** Reviewers ensure: 1) *Identity Consistency:* The same individual is consistent across clips for each speaker. 2) *Annotation Validation:* Automated annotations (2D keypoints, 3D SMPL-X, camera trajectories) are verified for a subset of samples. Anomalies are flagged for correction.

4. **General Quality Assessment:** Reviewers ensure: 1) *Speech Alignment:* The subject's lip movements align with the speech audio; 2) *Noise Filtering:* Clips with significant environmental noise or distractions are removed.

5. **Reviewer Training and Quality Audits:** To maintain consistency: 1) Reviewers are trained with examples of acceptable and unacceptable clips. 2) Periodic audits are conducted on random samples to ensure adherence to standards.

This multi-step process ensures that the dataset maintains high visual and audio quality, providing a robust foundation for research.

### A.4.2 DATA STATISTICS ON SHOT TYPES

Below are the results and corresponding analysis of the total number of clips and the distribution of shot sizes for each identity.

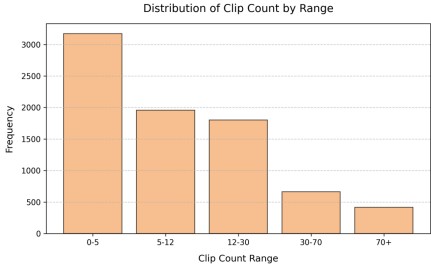 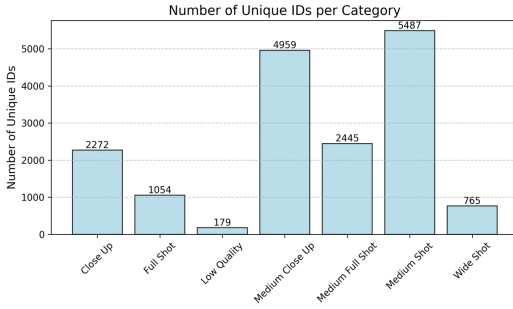

Figure 10: **Statistics of dataset clips: Left** - Clip Count Distribution per ID grouped by range. **Right** - Distribution of Unique IDs across Shot Categories.

Shown in Fig. 10, the bar chart in the left illustrates the frequency distribution of clip counts across predefined ranges. The X-axis represents different ranges of clip counts (0-5, 5-12, 12-30, 30-70, and 70+), while the Y-axis indicates the frequency, i.e., the frequency statistic represents the number of distinct clips associated with each ID across the entire dataset, falling within specific ranges. The Y-axis value corresponds to the number of IDs in each range. The bar chart on the right of

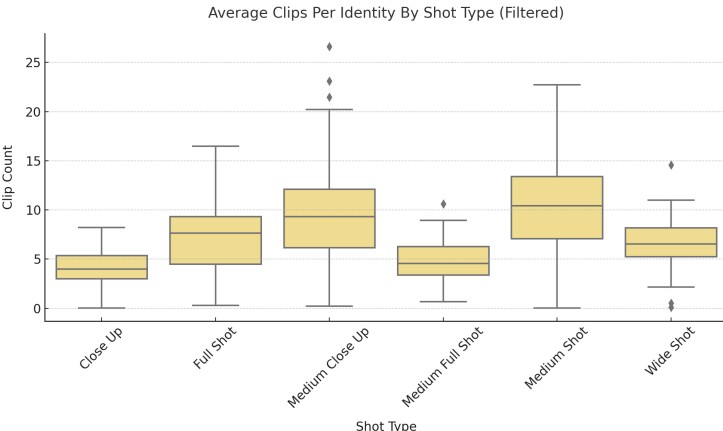

Figure 11: **Distribution of Average Clip Counts Per Identity Across Shot Types:** the boxplot shows the distribution of clip counts per identity across six shot types. The Y-axis represents the clip counts, and the X-axis categorizes the shot types. Each box represents the interquartile range (IQR), with the median as a horizontal line inside, whiskers indicating variability, and outliers shown as points.

Fig. 10 visualizes the number of unique IDs (identities) associated with each shot category. The X-axis represents the shot categories, including Close Up, Full Shot, Low Quality, Medium Close Up, Medium Full Shot, Medium Shot, and Wide Shot. The Y-axis shows the count of unique IDs for each category.

Additionally, shown in Fig. 11, each box represents the interquartile range (IQR), with the median as a horizontal line inside, whiskers indicating variability, and outliers shown as points. Medium Shot and Medium Close Up dominate with higher medians and broader distributions, while Full Shot and Wide Shot have lower medians and fewer outliers. This visualization highlights the variability and prevalence of shot types across identities.

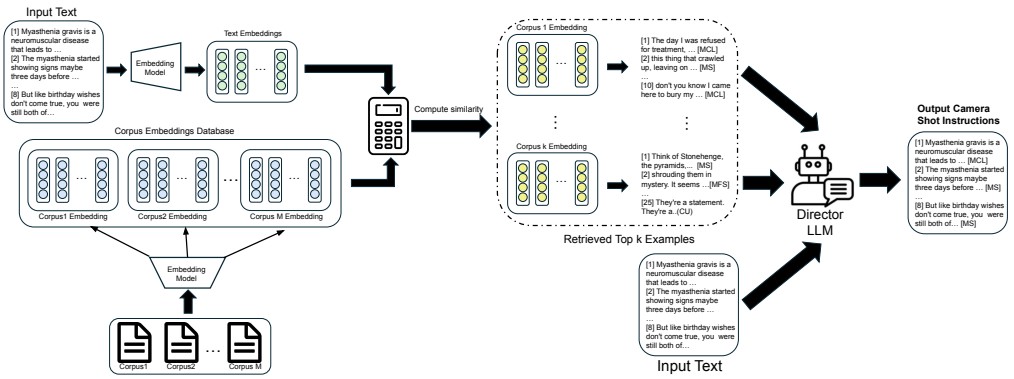

Figure 12: **Pipeline of RAG** in detail.

## A.5 DETAILS OF RETRIEVAL-AUGMENTED GENERATION

### A.5.1 RETRIEVAL-AUGMENTED GENERATION PROCESS

We provide a detailed illustration of the RAG process in Fig. 12. We aim to enhance the shot transition performance of LLMs using RAG. To achieve this, we use scripts with annotated shot transitions from the training dataset as the RAG corpus. The training dataset, composed of text scripts, is converted into an embedding dataset using the OpenAI text-embedding-small model.

For each input script, we similarly convert it into a text embedding using the same embedding model. Then, utilizing the FAISS (Douze et al., 2024) tool, we calculate the L2 distance between the input text embedding and each embedding in the embedding dataset. The top 5 files with the smallest distances are selected as the context, which is provided alongside the input script as input to the model.

### A.5.2 RETRIEVAL EXAMPLES

As shown in Figure 13, these are two examples of RAG assisting LLM in making shot transitions. In the first example, the **bold blue** portions of the script and relevant documents both express the love between a boy and a girl in a poetic manner. In the second example, the **bold blue** text highlight the importance of intimate relationships in helping humans confront pain and illness. The shot transition results in both examples align with those obtained through our RAG-based approach. The two examples respectively illustrate that the documents retrieved by RAG share similarities with our input scripts in terms of content or narrative logic. This demonstrates that the documents retrieved by RAG can indeed assist the LLM in making shot transitions.

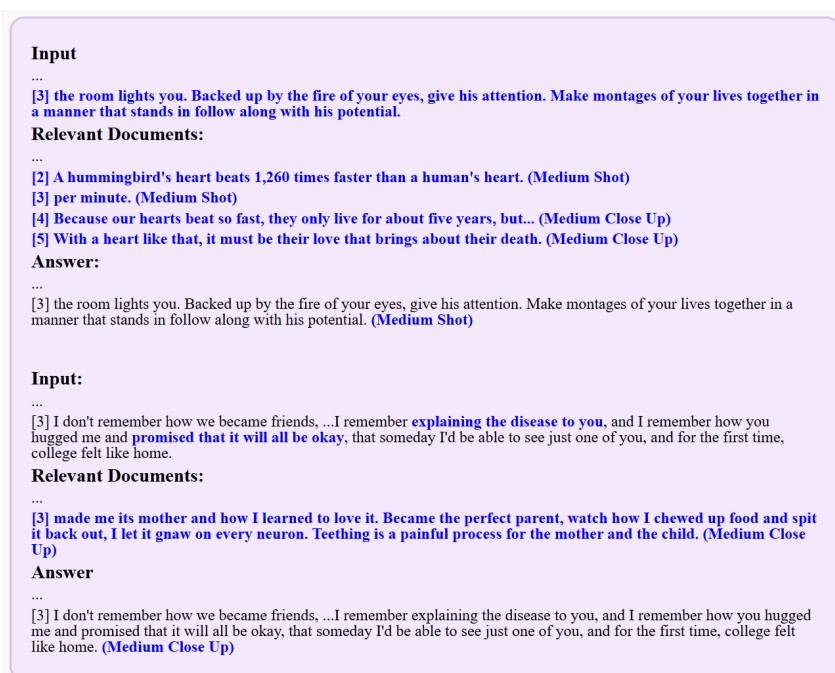

Figure 13: **Examples of RAG-Assisted Shot Transitions**:The **bold blue** text highlights similarities between the input script and the content retrieved from RAG documents.

### A.6 LIMITATIONS AND FUTURE WORKS.

Despite the effectiveness of our framework, several challenges remain unsolved. First, interaction with props and the environment (e.g., microphones or walking across a stage) is not yet seamlessly integrated into the generated videos, limiting the naturalness of the speaker's interaction with objects. Second, audience engagement elements such as eye contact, gaze shifts, and facial expressions are critical in talk shows and speeches but are difficult to capture and simulate without audience cues. Additionally, while our system handles multi-shot transitions effectively, it does not yet incorporate moving camera dynamics, which would further enhance the realism of the generated videos. As future work, we aim to explore moving camera integration leveraging advanced camera control modules.

## A.7 POTENTIAL PRACTICAL APPLICATION

The practical value of multi-shot speech video generation lies in its potential to revolutionize content creation across various industries by automating a traditionally labor-intensive and creative process. Key applications include:

- Entertainment and Media Production: This technology enables the efficient creation of dynamic, multi-shot speech videos for films, TV shows, and online content. By automating camera transitions, gesture synthesis, and vocal delivery, our system reduces the need for extensive manual editing and enhances the storytelling quality.
- Education: Multi-shot speech videos can be used to create engaging educational content, such as lectures or tutorials, where dynamic camera angles and gestures help maintain viewer interest and improve the conveyance of information.
- Corporate Communications: Businesses can use this technology to generate polished speech videos for presentations, product launches, or training sessions, offering a cost-effective way to produce professional-quality content.
- Content Creation for Social Media: Influencers and creators can leverage multi-shot speech video generation to produce compelling, visually engaging videos for platforms like YouTube, TikTok, or Instagram without requiring advanced editing skills or significant production resources.
- Virtual and Augmented Reality: Multi-shot speech videos could serve as a foundational component for immersive virtual presentations or augmented reality experiences, where dynamic and lifelike speech scenarios are crucial.

By addressing the complex challenge of generating long-form speech videos with dynamic camera shots, our work provides a foundation for these applications. The integration of the DirectorLLM with multimodal generation modules demonstrates a novel approach to orchestrating speech, motion, and visual elements in a cohesive manner. Our system reduces the barriers to high-quality video production, enabling creativity and innovation across industries. It offers a scalable solution that can adapt to various content requirements while maintaining consistency and realism. We believe that our research not only advances the technical capabilities in this domain but also opens up new possibilities for practical applications that can have a positive impact on entertainment, education, business, and more.

## A.8 POTENTIAL RISKS

Our proposed method presents risks related to potential misuse for misinformation campaigns and large-scale generation of fake news. To mitigate these concerns, we have carefully curated the dataset to include only innocuous topics such as education, entertainment, and public speaking in neutral settings. By focusing on benign subjects, we aim to minimize the potential for our work to be exploited for malicious purposes, while still demonstrating the effectiveness of our approach in a controlled and ethical manner. We are committed to responsible research practices and have taken deliberate steps to ensure that our contributions do not inadvertently contribute to the spread of misinformation or harmful content. Additionally, we encourage further exploration of ethical safeguards and detection mechanisms to prevent misuse.

