# OpenReview forum: "Orator: LLM-Guided Multi-Shot Speech Video Generation"
_ICLR.cc/2025/Conference — Submitted to ICLR 2025_

### Official Review · Reviewer_J3eb · 2024-10-30

**Soundness:** 3
**Presentation:** 3
**Contribution:** 3
**Rating:** 8
**Confidence:** 4

**Summary:**

In this method, the user needs to input a reference set containing the desired shots and text lines. A DirectorLLM then serves as a multi-role director, controlling the motion, camera transitions, and audio of the generated video. The authors also introduce a new dataset, TalkCuts, to support community research. The experimental results show promising outcomes.

**Strengths:**

1. They utilize a DirectorLLM to control the camera transition, motion, and audios. In this way, the story of generated video should be more realistic compared with mannually control.
2. They contribute a more powerful dataset TalkCut.
2. They show promising results in experiment part.

**Weaknesses:**

1. The temporal consistency of  results in their project page needs improvement.
2. It seems that SMPL-X doesn’t model the mouth region, which makes that part look unnatural.
3. The Retrieval-based Augmentation Generation in Figure 3 is unclear.

**Questions:**

1. In Figure 1, do you need to provide a reference image for each shot? For example, is it because the reference image you input shows a close-up shot, and DirectorLLM then guides the generation of a close-up clip based on that reference? If possible for Orator to produce novel-view clips? For example, given several full body reference images, the Orator generates a close-up shot clip.
2. If possible, the text lines can also be remove, just give a story brief and ask the DirectorLLM to generate the text lines according to the given story. More than that, DirectorLLM can also generate the description of the reference image by story and use a SD to generate reference images.
3. This system looks interesting. However, it gives me a feeling that this paper is collect those existing SpeechGen, MotionGen, VideoGen models and utilize DirectorLLM to control them.
4. I still love this paper, I would like upgrade my rating if author can give me more insight during rebuttal.

---

### Official Review · Reviewer_A2Pw · 2024-10-30

**Soundness:** 3
**Presentation:** 2
**Contribution:** 3
**Rating:** 3
**Confidence:** 4

**Summary:**

This paper introduces Orator, a system for generating multi-shot speech videos with natural camera transitions. This paper introduces a large-scale dataset featuring over 500 hours of speech videos with multi-angle shots, 3D motion annotations, and camera trajectories.
On top of that, Orator leverages a large language model (LLM) as a “director” to generate detailed instructions for camera transitions, gestures, and vocal delivery, which guides a multi-modal video generation module to produce coherent, long-form speech videos.

**Strengths:**

1. The paper explores the problem of how to generate multi-camera speech videos with dynamic lens transitions, which extends a previous method for generating single-camera half-body videos.
2. The method is well supported by the introduction of TalkCuts. This dataset combined with a well-designed multimodal framework can generate multicamera speech videos with dynamic view transitions.

**Weaknesses:**

1. The data collection and annotation process is fully automated, with no manual verification involved. This raises concerns about the dataset's quality.

2. I don’t see clear evidence of text similarity between the input speech scripts and the corpus that could be used for transition planning.

3. In Lines 192-193, you mentioned that "each identity is recorded with multiple diverse camera shots." Could you provide statistics on the number and type of shots for each identity, particularly for “volg”?

4. Please share the access link to your dataset. The model appears to have numerous modules, and without access to the code, it will be challenging to reproduce your results.

5. L364-365: typo.

**Questions:**

What is the practical value and application of the multi-shot speech video generation？

---

### Official Review · Reviewer_8WEn · 2024-11-02

**Soundness:** 3
**Presentation:** 2
**Contribution:** 3
**Rating:** 5
**Confidence:** 4

**Summary:**

This paper aims to achieve end-to-end generation of speech videos by combining a large language model (LLM) and a diffusion model. The input only needs to contain speech text and pictures of different shots, and the final output is a speech video (with shot switching). By combining a series of existing methods, the author presents some generated results and verifies the feasibility of the task.

**Strengths:**

- The task that the author hopes to complete is quite interesting and also very difficult. The generated results provided to some extent illustrate the feasibility of the task.
 - The video generation part implemented by the author has certain advantages in the synthesis of hand details compared with some existing methods.

**Weaknesses:**

- The overall effect is not good. In the synthesized final speech video, there are obvious artifacts in the lip-sync accuracy and the identity change when switching shots. In the last two examples on the homepage, the audio and lip movements are poorly synchronized, with many syllables lacking corresponding mouth movements. Additionally, there are issues with maintaining appearance consistency, such as blurry hair.
 - The novelty is limited. The video generation part is implemented by combining SVD and ControlNet. It is within expectations that better results can be achieved by training on speech video data. Perhaps a more end-to-end model design would be more innovative. For example, removing SMPL as an intermediate representation and eliminating explicit shot state representations could reduce information loss during intermediate transmission and produce more natural results.
 - There is a lack of any overall effect evaluation. The author only made performance comparisons of the results for each sub-module. However, the comparisons of video generation and speech to gesture generation are both cases where the fine-tuned model has performance advantages over the baseline. This is very normal and not a difficult thing.

**Questions:**

(1) In the two complete generated results displayed on the homepage, are all the IDs in the source images untrained? That is, is it a completely one-shot setting? I haven't found an explanation for this. If it is omitted, please let me know. Thanks.

(2) I haven't found an evaluation of the final generated video, which is the purpose of this paper. There are many evaluations of specific sub-modules, but there is not much novelty contribution in terms of method design. The evaluation method of the final generated video may also be an important content of this paper and should be discussed in detail and be self-contained.

(3) For the evaluation of the final generated video, although there is a lack of directly comparable methods, I believe there are two key aspects that need to be assessed. These aspects determine whether the proposed method in this paper is (1) a stitched-together approach that combines existing methods, or (2) a functional framework prototype. Firstly, regarding the quality of the synthesis, it can be evaluated using objective metrics such as PSNR and pretrained video quality assessment models comparing with real speech videos, as well as subjective scoring to compare the current performance gap. Secondly, concerning the usability rate of the synthesis results, we should assess how many of the randomly generated results do not exhibit severe artifacts (which can be considered completely non-functional), such as body deformities, meaningless shot cuts, and extremely low lip-sync accuracy. For the proposed speech video generation task, currently having only two video results is insufficient.

Overall, this paper addresses a very interesting and highly challenging task. However, the experimental section is not sufficiently comprehensive. It lacks extensive experimental results to validate the effectiveness of our approach in generating engaging and realistic multi-shot speech videos. Instead, it primarily demonstrates improvements of individual components over baselines in their respective tasks. There is no detailed experimentation on whether the approach can effectively and holistically generate engaging and realistic multi-shot speech videos. This additional validation will be very important.

---

### Official Review · Reviewer_b2gi · 2024-11-02

**Soundness:** 3
**Presentation:** 2
**Contribution:** 2
**Rating:** 5
**Confidence:** 2

**Summary:**

The innovations of this paper include the Orator system and the TalkCuts dataset: the multimodal video generation module in the Orator system integrates the collaboration of multiple sub-modules, each with its own unique strengths, and the DirectorLLM acts as a multi-role director to guide the video generation; the TalkCuts dataset is characterized by its large scale, diversity, and rich annotation information, which provides a powerful support to multicamera speech video generation. The TalkCuts dataset is characterized by large scale, diversity and rich annotation information, which provides strong support for multi-camera speech video generation.

**Strengths:**

1. By integrating the multimodal video generation module and DirectorLLM, the Orator system achieves effective coordination and fine control of the video generation process, providing a new solution for multicamera speech video generation.
2. The TalkCuts dataset is large in size, rich in diversity and comprehensively labeled, which fills the gap of existing datasets in multi-camera speech video generation and provides important data support for the research.
3. Comprehensive experimental evaluations are conducted on several key tasks, with improvements over the baseline model, demonstrating the effectiveness and sophistication of the method.

**Weaknesses:**

1.the table does not have the best results in bold, I hope to pay attention to the details.
2.Although the experimental results show some advantages over the baseline model in some metrics, the demo shows that there is a noticeable abruptness when switching between shots as well as the human body does not have a good timing consistency
3.The dataset is primarily focused on the speech domain, which may limit the generalizability of the model, and may have more application scenarios in other domains such as movies.

**Questions:**

1. I checked your online demo and noticed that your method has a noticeable effect on the background for the body, similar to a change in light and shadow? What is the reason for this?
2. For the model combining Stable Video Diffusion and ControlNeXt in the VideoGen module, how to balance the contribution of the two models during the training process to achieve the best video generation results?
3.When there are ambiguities or conflicts in the instructions given by DirectorLLM, how does the system handle them to ensure coherent and reasonable video generation?
4. For the large number of videos collected, what are the specific screening criteria and process in the manual screening process? How to ensure the consistency and accuracy of screening?


Generated videos were not seamlessly integrated in terms of interaction with props and environments, such as the lack of speaker interaction with the microphone or walking on stage, limiting the naturalness of the videos.
Noted that audience engagement elements such as eye contact, gaze shifts, and facial expressions were difficult to capture and simulate because of the lack of audience cues. Although the system can handle multi-camera transitions, it has not yet incorporated moving camera dynamics, which affects the realism of the video.

---

### Official Review · Reviewer_uS7p · 2024-11-04

**Soundness:** 3
**Presentation:** 3
**Contribution:** 3
**Rating:** 6
**Confidence:** 2

**Summary:**

This work is technically novel and interesting. The proposed Orator system adopts the LLM guided multimodal generation framework, which can automatically coordinate the camera transitions, speaker gestures and voice outputs to generate coherent and attractive multicamera speech videos. At the same time, they also created a new large-scale dataset, TalkCuts, which contains hundreds of hours of richly annotated multicamera speech videos, which is useful for research in related fields. However, the writing and experimental parts of the paper still need some improvement.

**Strengths:**

A new task is proposed: speech video generation with dynamic camera switching. The first large-scale dataset dedicated to this task, TalkCuts, was created, and a novel multimodal generation framework, Orator, was proposed, in which DirectorLLM acts as a multi-role director to guide the process. These are all highly original contributions.

**Weaknesses:**

Generally speaking, the paper is understandable, but some details are not clear enough, for example, the mechanism of how DirectorLLM directs the work of each module could be explained in more detail. The experimental part also lacks more quantitative and qualitative results to support the claimed advantages.

**Questions:**

The overall methodology is reasonable, but there is still a lack of justification in some details, such as the lack of more experimental results to quantitatively assess the metrics of authenticity, coherence, and diversity of the generated videos.

---

### Meta-Review · Area_Chair_Xreg · 2024-12-19

**Metareview:**

This paper presents an Orator system that adopts the LLM guided multimodal generation framework, which can automatically coordinate the camera transitions, speaker gestures and voice outputs to generate multi-camera speech videos. A new large-scale dataset, TalkCuts, is provided which contains annotated multi-camera speech videos.

However, reviewers expressed concerns on the overstatement of the paper and the visual results.
Some obvious limitations have been pointed out by the reviewers, such as the requirement to provide a reference image for each camera shot. Furthermore, reviewers b2gi, 8WEn, and J3eb all spotted the demo issues. Due to these issues, the rebuttal did not change the mind of reviewers though the writing has improved during revision.

AC made the decision due to the obvious limitations of this work and noticeable artifacts in the results.

**Additional Comments On Reviewer Discussion:**

This paper received mixed reviews. After the rebuttal, reviewers have not changed their mind due to the weakness pointed out by reviewers. In particular, reviewers have cross-check others' comments.

---

### Decision · Program_Chairs · 2025-01-22

Reject